# Publication Trends in Neglected Tropical Diseases of Latin America and the Caribbean: A Bibliometric Analysis

**DOI:** 10.3390/pathogens10030356

**Published:** 2021-03-17

**Authors:** Gustavo Fontecha, Ana Sánchez, Bryan Ortiz

**Affiliations:** 1Microbiology Research Institute, Universidad Nacional Autónoma de Honduras, J1 Building, 4th fl, Tegucigalpa 11101, Honduras; bryanortiz_02@hotmail.com; 2Department Health Sciences, Brock University, 500 Glenridge Avenue, St. Catharines, ON L2S 3A1, Canada; asanchez@brocku.ca

**Keywords:** neglected tropical diseases, Latin America and the Caribbean, bibliometric analysis, HIV/AIDS, malaria, tuberculosis

## Abstract

(1) Background: Neglected tropical diseases (NTDs) have been overlooked on the global health agenda and in the priorities of national systems in low- and middle-income countries (LMICs). In 2012, the Sustainable Development Goals (SDGs) were created to ensure healthy lives and promoting well-being for all. This roadmap set out to accelerate work to overcome the global impact of NTDs. Almost a decade has passed since NTDs were re-launched as a global priority. Investment in research and development, as well as the production of scientific literature on NTDs, is expected to have increased significantly. (2) Methods: A bibliometric analysis of the scientific production of Latin America and the Caribbean (LAC) was carried out in relation to 19 endemic NTDs. These data were compared with the scientific production in malaria, tuberculosis, and HIV/AIDS. The database available from Thomson Reuters Web of Science (WoS) was used. In addition, the average annual growth percentage was calculated for each disease. (3) Results: In the last decade, the NTDs with the highest number of publications in the world were dengue and leishmaniasis. The United States was the most prolific country in the world in 15 out of 19 NTDs analyzed. In the LAC region, Brazil was the largest contributor for 16 of the 19 NTDs analyzed. Arboviral diseases showed the highest average annual growth. The number of publications for malaria, tuberculosis and HIV/AIDS was considerably higher than for NTDs. The contribution of most LAC countries, especially those considered to be LMICs, is inadequate and does not reflect the relevance of NTDs for the public health of the population. (4) Conclusions: This is the first bibliometric analysis to assess the trend of scientific documents on endemic NTDs in LAC. Our results could be used by decision makers both to strengthen investment policies in research and development in NTDs.

## 1. Introduction

Infectious diseases have been one of the leading causes of human morbidity and mortality since the origins of our species, and some of them have shaped our evolution and continue to affect our lives [1,2]. Currently, communicable diseases continue to affect all geographic areas; however their burden is greater in developing nations considered to be low- and middle-income countries (LMICs) [3]. In the last two decades, diseases that mainly affect the tropical and subtropical regions of the planet have received the attention of the World Health Organization, as well as from other stakeholders such as the scientific community, national leaders, non-governmental organizations and funders [4,5]. 

A group of diseases affecting impoverished urban and rural populations was initially named as “Neglected Tropical Diseases” (NTDs) by a group of scientists and activists. One of the NTDs common factor is that they almost exclusively affect populations living in extreme poverty in very remote geographic areas, or socially marginalized [4,6]. WHO initially recognized a group of 13 tropical infections as part of the official priority list of NTDs, but now the list has been expanded to a set of 20 diseases and disease groups [7]. Among the community of NTD investigators, however, attention is given to more than 42 groups of infectious diseases and a couple of non-infectious conditions such as the snakebite envenoming [8]. 

The NTD designation is as useful today as it was 20 years ago to underscore the importance of many tropical diseases that had been included in the vague category of "other diseases" in the Millennium Development Goal (MDGs), specifically MDG number 6 [9]. The MDGs focused primarily on the so-called “big three” killers—AIDS, malaria, and tuberculosis (TB) [10]—although the collective burden of disease from NTDs was —and still is—higher in some regions of the world, such as Latin America and the Caribbean (LAC) [11,12]. The deadline for achieving the MDGs has now ended, giving way to the reformulated Sustainable Development Goals (SDGs), which in SDG number 3 propose “*to end the epidemics of AIDS, tuberculosis, malaria and NTDs and to combat hepatitis, water-borne diseases and other communicable diseases*” by 2030 [13]. The SDGs are a global showcase that draws renewed attention over NTDs and are a historic opportunity to allocate funds for research and development [14]. 

Based on the NTDs roadmap published by WHO in 2012 [15] and 2013 [16], scientific research on these topics is experiencing an increasing trend [5,17,18,19]. Analyzing bibliometric data is a powerful approach for revealing research efforts to inform policy and practice as well as to support decisions to strengthen research capacities, particularly in LMICs. Bibliometric analysis is also an important indicator of the scope of national and or international initiatives to combat the roots and consequences of poverty (such as NTDs) [20,21]. The analysis of scientific production contained in the bibliographic databases makes it possible to highlight countries´ investment gaps in priority issues for their inhabitants. This in turn can improve the uptake of evidence for health actions [22]. This is particularly relevant in LAC, the region of the world with the highest inequalities anywhere [23] and one significantly affected by tropical diseases [11]. 

It is in this context that we carry out a 10-year bibliometric analysis of NTD research in LAC, with the objectives of (*i*) describing the region´s contribution to scientific literature in comparison with the rest of the world; (*ii*) comparing the scientific output of 19 endemic NTDs and the “big-three” infectious diseases: malaria, TB and HIV/AIDS, and (*iii*) assess the scientific contribution among countries in the Americas.

## 2. Results

### 2.1. Publication Output

The bibliographic search on the 19 NTDs described above, yielded a total of 266,846 publications worldwide for the period 2010-2019. The largest number of publications were on dengue (*n* = 18,186), leishmaniasis (*n* = 16,589), and Chagas disease (*n* = 9253) (Table 1). On the other hand, the three NTDs with the lowest number of publications were yaws (*n* = 111), chromomycosis (*n* = 365) and mycetoma (*n* = 451). The overall number of publications for the so-called “big three” infectious diseases was considerably higher than for the NTDs: TB (*n* = 70,139), malaria (*n* = 48,484), and HIV/AIDS (*n* = 46,293).

### 2.2. Most Productive Countries

The United States had the largest output of scientific publications in the world for 15 out of 19 NTDs analyzed (Table 1). USA was also the country with the largest number of scientific publications on malaria, TB, and HIV/AIDS. However, Brazil was the largest contributor on Chagas disease, chromomycosis, leishmaniasis and leprosy (Table 1).

In the LAC region, Brazil was the largest contributor for 16 out of 19 NTDs analyzed, whereas Mexico was the country with the greatest number of articles regarding taeniasis/cysticercosis and mycetoma. Argentina was the most prolific country for hydatid disease publications.

### 2.3. Most Productive Journals, Languages, Document Types, Most Prolific Authors and Institutions

The journal with most publications in the field of NTDs was *Plos Neglected Tropical Diseases* (for ten out of 19 NTDs), followed by the *American Journal of Tropical Medicine and Hygiene* (5 out of 19 NTDs). Of the three most commonly spoken languages in the Americas, retrieved publications were written mainly in English (98.46%), while 1.05% were published in Spanish and 0.49% in Portuguese.

With respect to the type of document, research articles were predominant (75.23%), followed by reviews (9.96%), meeting abstracts (8.95%), editorial material (3.15%) and letters (2.72%). The top productive authors and institutions for each topic are listed in Table 1. 

When considering the 19 NTDs as research topics, the University of Sao Paulo in Brazil was the most productive institution for Chagas disease, Leishmaniasis and Leprosy, while the Swiss Institute of Public and Tropical Health ranked first for publications on Schistosomiasis and Soil-Transmitted Helminthiases, The US Centers for Disease Control and Prevention (CDC) contributed with the largest number of publications for Zika and Rabies. In terms of authors’ affiliations, institutions in the United States and Switzerland had the highest output for five and four of the 19 NTDs, respectively. The analysis by type of institution, results indicated that for 13 NTDs most productive authors were affiliated with universities, whereas the rest were hospitals, public health institutes, research centers, or foundations.

### 2.4. Annual Publication Trends

The topics with the highest average annual growth between 2010 and 2019 were the arboviral infections: Chikungunya (*n* = 11.98%), dengue (*n* = 8.75%) and zika (8.41%). These three topics had an average annual growth even higher than TB (5.73%), malaria (2.29%), and HIV/AIDS (−0.82%). The topics with the highest average annual decrease were yaws (*n* = −8.24%), mycetoma (*n* = −2.45%) and chromomycosis (*n* = −1.18%) (Figure 1A). However, the average yearly number of publications for any of the "big-three" was significantly higher than for any of the NTDs analyzed (Figure 1B).

### 2.5. Comparing Scientific Productivity between Countries

As shown in Figure 2 (more extensive data are also showed in Table 2), the LAC region contributed more scientific publications than the US and Canada for eight NTDs (Chagas, chromomycosis, cysticercosis, hydatid disease, fascioliasis, leishmaniasis, leprosy, and mycetoma). The topic in which LAC contributed the most to world knowledge was Chagas disease (56.67%), while the opposite was true for buruli ulcer (2.08%). A complete list showing the number of publications by disease and by country of the LAC region can be found in Appendix A.

The two topics for which the Americas as a continent contributes the most to the world are zika (78.26%) and Chagas (74.19%). The NTDs in which the Americas as a continent contribute more than 50% of the world’s publications were: Chagas, chromomycosis, cysticercosis, dengue, zika, chikungunya, leishmaniasis, leprosy, onchocerciasis, and soil-transmitted helminth infections (STH). The Americas is also the region with more contributions for HIV/AIDS.

Figure 3 depicts the marked heterogeneity that exists between the number of scientific publications worldwide for the “big-three” diseases (malaria, TB and HIV/AIDS), with respect to NTDs when they are separated by taxonomic categories (parasitic diseases, bacterial and fungal diseases, and viral diseases).

### 2.6. Occurrence Map of the Most Frequent Keywords 

The network of the most frequently mentioned terms in publications under the topic of “neglected tropical disease” is presented in Figure 4. The figure consists of five clusters of associated terms in different colors. The blue cluster included terms such as cutaneous leishmaniasis, expression and resistance. The red cluster included terms such as in vitro, leishmaniasis, drug discovery and identification, and the green cluster highlights terms such as NTD, prevalence, diagnosis, and epidemiology.

## 3. Discussion

We conducted a comprehensive bibliometric research on LAC´s scientific production in relation to the Americas and the world in the field of NTDs. The “NTD brand” was coined at the early 2000s to name a group of diseases that had been left out of the political scenario when the Millennium Development Goals (MDGs) were formulated [24]. Despite the fact that this already long list of diseases has received relatively little attention from international organizations and national governments compared to the so-called "big three” killers—TB, malaria, and HIV/AIDS—there are important aspects of how some diseases within the “NTD family” are prioritized. According to our results, there are marked differences in the number of publications on diseases such as dengue, leishmaniasis and Chagas disease, and the number of articles on yaws, chromomycosis and mycetoma (up to 150 times more). From a scientific perspective, it appears that some NTDs have received more attention than others. This could be due to differential burden of disease between countries, geographic distribution, and available funding from governments and funding agencies. Whatever the reasons behind the differences between NTDs research output, our results highlight the need for additional investment and further science for some of the most neglected diseases in the list.

One of the aims of this bibliometric study was to highlight the indisputable difference in scientific output between the “big-three” diseases and the NTDs in the last decade. In recent years, NTDs have gained visibility and have been explicitly included in the SDGs, under the beautiful motto “*leaving no one behind*” [25]. Most recently, the World Health Assembly approved the new and ambitious roadmap targets for NTDs for 2030 [7]. This change has led to an increase in publications related to NTDs in the scientific literature [5,26] but not enough to surpass TB, malaria and HIV/AIDS, as our findings demonstrate. With the relaunch of NTDs in the SDGs, international organizations and local governments are expected to encourage investigators to conduct research activities in all NTDs aspects, especially applied and operational research. We hope that the deserved attention given to the “big-three killers” does not eclipse the suffering that NTDs keep causing in the most disadvantaged populations of the planet [10].

This study also set out to describe LAC´s scientific contribution in relation to NTDs at the global level, and to compare the contribution of different countries in the region. As is the case for the vast majority of knowledge areas, the United States is the largest contributor for 15 of the 19 NTDs analyzed, and is also the most prolific country with regards to malaria, TB, and HIV/AIDS. These results are not surprising and confirm that authors from non-tropical high-income countries where NTDs are not major public health problems often collaborate with authors in the Global South [5,27]. 

Historically, these collaborations, as noted by Palmblad and Torvik 2017 may stem from shared colonial history and language [19]. In the present times, incipient research development in many tropical countries drives the need for scientific collaboration with more developed nations. In addition, research outputs in this field maybe reflect common interest related to geopolitical circumstances. An example of the latter is the fact that the U. S. Army and the U. S. Navy deploy temporary or permanent units to developing countries (e.g., in the Americas: Peru, Colombia, and Honduras, among others) to participate in peacekeeping operations, joint military exercises, and establish collaborations on epidemiological surveillance with local health or academics institutions [28]. Since military personnel can either spread infectious diseases within a population, or become at risk of infection by pathogens endemic to the host tropical countries, collaborative biomedical and epidemiological research becomes indispensable to address infectious diseases risks.

Our findings show the outstanding contribution of Brazil worldwide, especially in areas such as Chagas disease, chromomycosis, leishmaniasis and leprosy. In fact, Brazil is the main contributor from LAC in 16 of the 19 NTDs analyzed in this study. Other bibliometric analyses also show Brazil as one of the main contributors in publications related to infectious diseases in general [3,5,26,29,30]. These data place Brazil as the leader in tropical medicine in the Latin American region and are directly related to the heavy burden that communicable diseases continue to exert among the Brazilian population [31] and the budget effort by the Brazilian government [32]. 

In this study, the regional contribution of LAC as a whole was greater than that of the United States and Canada in eight topics: Chagas disease, chromomycosis, taeniasis/cysticercosis, hydatid disease, fascioliasis, leishmaniasis, leprosy and mycetoma. The contribution of Mexico on taeniasis/cysticercosis and mycetoma, and that of Argentina on hydatid disease stand out. In both cases, these research outputs are a reflection of particular local health needs of each country. This is also the case of Zika, a disease for which the American continent has contributed more than 78% of the publications worldwide in the last decade, due to the high impact of the recent epidemic in the region since 2015 and the dire consequences for the health of newborns [33,34].

Now, if we exclude Brazil from the analysis, most of the countries in LAC lag behind other developing regions, in contrast to the high prevalence rates of many NTDs [12]. Small countries in Central America and the Caribbean are perceived as under-productive in the NTD field [35,36], particularly considering that their collective burden may exceed other conditions such as HIV/AIDS, TB or malaria [11]. Forty-nine of the 55 countries (89%) of the continent contribute less than 1% of publications on NTDs to the scientific literature, and 34 of those countries (62%) contribute less than 0.01% of the world total (Appendix A). This imbalance confirms that the lowest-income countries with the highest disease burden are not contributing adequately to the field of tropical diseases [3], and reinforces the paradigm of the “10/90 gap” [37,38], which establishes that less than 10% of global funding for research is spent on diseases that afflict more than 90% of the world´s population [39]. These data highlight the need to develop and support research capacity at the individual and institutional level in the most underdeveloped and endemic countries [40]. Local research capacities can and should be strengthened through investment in research facilities and training programs for young researchers. 

In this regard, it is evident that the lower-income countries of LAC such as the so-called “northern triangle of Central America” (Guatemala, Honduras, and El Salvador) have limited resources to invest in health research and they have historically depended on external cooperation. According to bibliometric results, donor funding is not enough to achieve the desired sustainability in scientific research. Choi et al. 2015 have introduced the term “self-neglect” to refer to the lack of support from governments and the apathy of victims of tropical diseases [41]. These authors suggest that “self-neglect by victims is the core of the neglect”. They properly suggest that as victims demand greater attention and take ownership of the problem, prevention efforts are more effective.

One way to show that governments and policy makers in LAC LMICs must reorient the distribution of national resources toward the real priorities of their inhabitants is to compare the percentage of defense spending relative to gross domestic product (GDP) (Figure 5) [32,42]. In this figure, some countries of the American continent have been classified into three groups: (a) countries that invest a higher percentage in research and development (R&D) than in military spending such as Canada and Costa Rica; (b) countries with a balance between R&D and military spending similar to the world average, such as Argentina and Brazil, and (c) countries with military spending disproportionately higher than investment in R&D. Ironically, lower-income countries such as Honduras and El Salvador have the highest budgets dedicated to military defense and the lowest budgets for science. These data confirm the concept of “self-neglect” applied to the governments of the countries with the highest burden of disease and also point to the geopolitical history of these small nations.

The route to achieve the SDGs by 2030 has been well outlined by the WHO. Some countries in the region have assumed their historical responsibility toward their citizens; however, other countries, particularly communities and national governments, need additional investment and an intensified effort to achieve health goals in relation to NTDs [25].

## 4. Materials and Methods

### 4.1. Database

In the current study, a bibliometric approach was implemented to generate qualified information related to 19 NTDs endemic in Latin America and the Caribbean and the so-called “big-three” infectious diseases (malaria, TB, and HIV/AIDS) (Table 1). Scientific publications indexed in Thom­son Reuters Web of Science (WoS) Core Collection [43] were included in the study. 

### 4.2. Search Strategy

The search was car­ried out in October 2020, encompassing the years of 2010 to 2020. The advanced search mode was selected using the title (TI) and abstract (AB) search fields. The search queries used were: [TI = (MeSH)] OR [AB = (MeSH)]. For most of the 22 diseases, two or more Medical Subject Heading (MeSH) Terms were searched as shown in Table 1, including the scientific name of the pathogen and the most common name of the disease. For rabies and HIV/AIDS the Boolean operator was “AND” instead of “OR” to avoid false positive results. Data retrieved from each query was visually validated to prevent false positive results as a consequence of multiple meanings of the terms. Only English search terms were used because, in general, terms in other languages such as Spanish or Portuguese do not significantly increase the results.

Additionally, to determine the most frequent terms in the last 500 publications included under the topic [TS = neglected tropical diseases] a bibliometric map was elaborated using the VOSviewer software v. 1.6.15 (https://www.vosviewer.com (accessed on 15 January 2021)).

### 4.3. Bibliometric Parameters

All types of publications were included in each search. The following parameters were selected for further analysis and interpretation of results: (i) number of documents worldwide, (ii) number of documents for each American country, (iii) countries with the highest number of publications, (iv) LAC country with the highest number of publications, (v) institutions with the highest number of contributions, (vi) the most common language of the publications (English, Spanish or Portuguese), (vii) the most frequent source (journal), (viii) yearly publications, and (ix) author with most publications per NTD and his or her and affiliation. The following types of document were further analyzed according to their percentage of the total: original articles, reviews, meeting abstracts, editorial material, and letters. As in any other bibliometric analysis, the results of the WoS search include publications whose authors may belong to several institutions and to more than one country. Consequently, in many cases the results are a reflection of the collaboration of LAC scientists with countries in other regions, or of scientists from two or more LAC countries.

### 4.4. Data Analysis

Yearly publications for each disease were recorded to calculate the average annual percentage growth (AAPG) from 2010 to 2019. Documents published in 2020 were excluded from this analysis in order to only compare the years that had a complete collection of records. 

For each year from 2010 to 2019, “Z” was calculated as the number of publications in one year subtracted from the number of publications in the previous year:Z = [N ^year A^ − N ^year A−1^]
where N is the number of publications.

Then the annual percentage growth (APG) was calculated by the difference of this subtraction divided by the total publications of the last year multiplied by 100:APG = [Z/N ^year A^] × 100

Then, the average annual percentage growth (AAPG) was calculated by the average of the nine data:(AAPG) = AVERAGE [APG1 + APG2 + … APG9]

To calculate the average number of publications (ANP) per year for each disease, the number of publications from 2010 to 2019 was added and divided by 10.

The number and percentage of publications produced in LAC and in the continent as a whole were also calculated for each disease (Appendix A) and compared with respect to the total number of publications in the world. Finally, the number of publications on NTDs of parasitic etiology was compared separately with the number of publications of malaria. Similarly, the number of publications on bacterial and fungal NTDs was compared with those of TB, and the number of publications on viral NTDs was compared with those on HIV/AIDS.

Data were tabulated and processed in Microsoft^®^ Excel for Mac (version 16.42, Microsoft 2020), and plotted in Graphic for Mac (version 3.1, Picta Inc. 2018).

## 5. Conclusions

This study is the first bibliometric analysis assessing the trend of published documents regarding the most common NTDs and the “big-three” infectious diseases in LAC region, and it can assist researchers as well as national and international policy makers in guiding, planning, and funding decisions in order to achieve the SDGs for global health.

## 6. Limitations

We restricted our bibliometric analysis to Web of Science (WoS) database, and we did not compare our findings with other homologous databases such as Scopus, PubMed, ScIELO, Cochrane Library, or Embase, which could help provide a better overview of the published literature in the field of NTDs. Another limitation of the study is that it did not include several NTDs relevant to LAC such as snakebite envenoming, strongyloidiasis, leptospirosis, yellow fever, trichinosis, among others. We also recognize that in any bibliometric study, the selection of the databases and especially the selection of the MeSH terms are a source of bias in the results obtained. Consequently, the results presented are only a reflection of reality.

## Figures and Tables

**Figure 1 pathogens-10-00356-f001:**
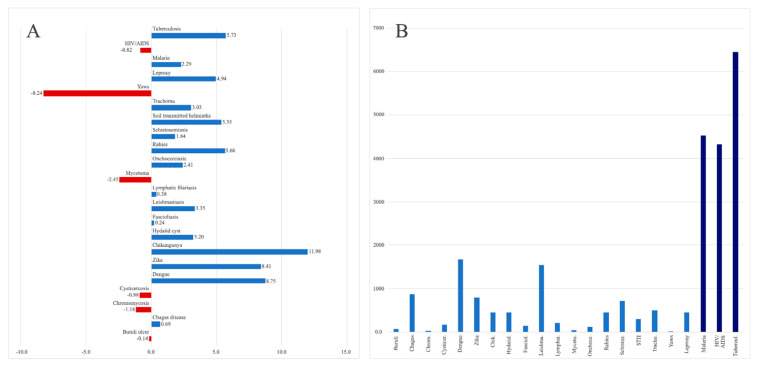
(**A**) Average annual percentage growth (AAPG) of scientific publications from 2010 to 2019, (**B**) Average number of scientific publications (ANP) per year from 2010 to 2019.

**Figure 2 pathogens-10-00356-f002:**
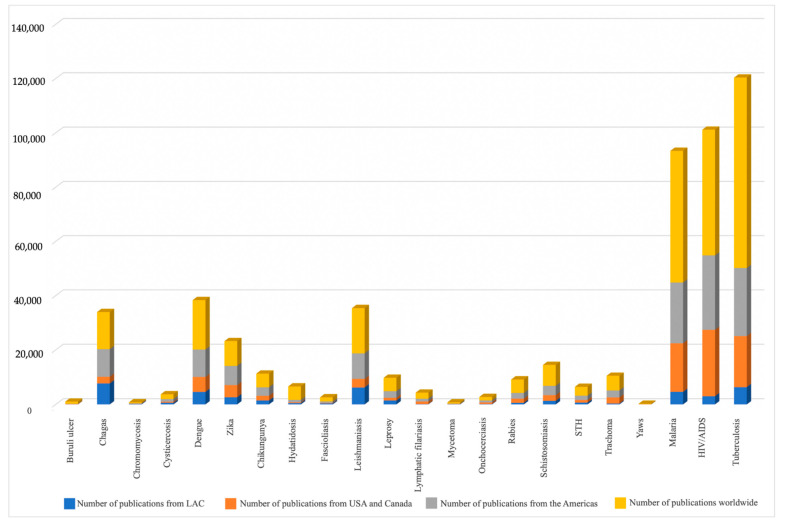
Number of scientific publications by topic in Latin America and the Caribbean, the Americas and in the world, between 2010 and 2019.

**Figure 3 pathogens-10-00356-f003:**
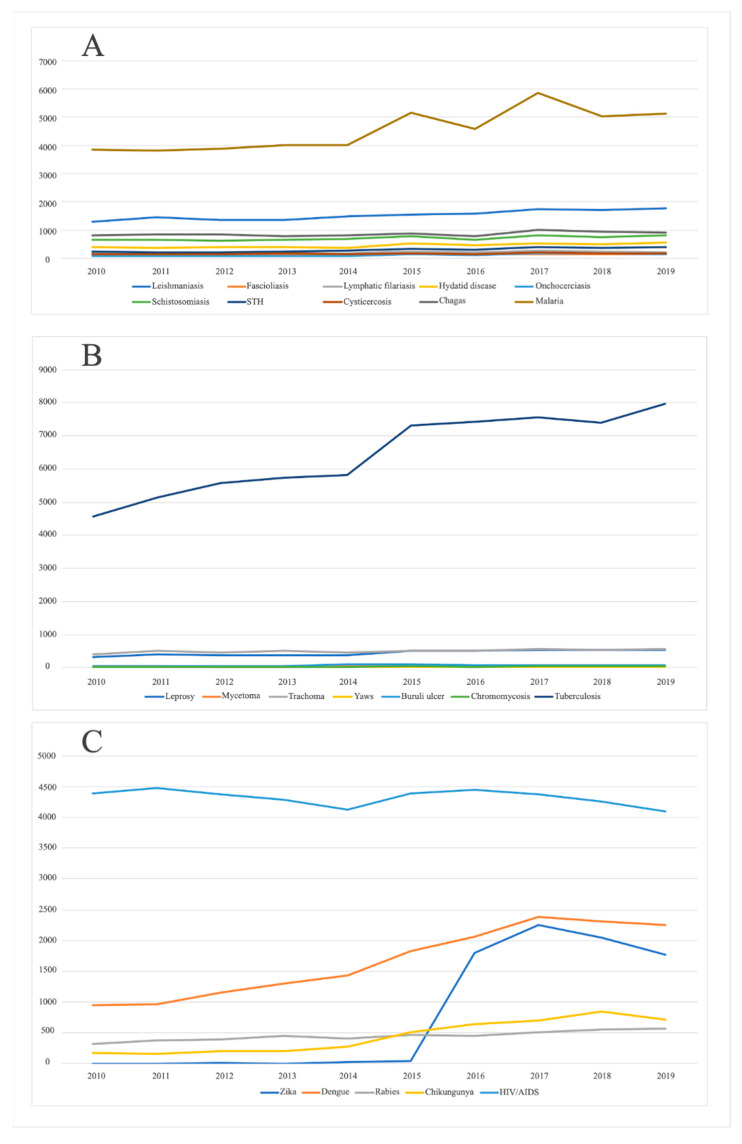
Number of publications per year between 2010 and 2019 according to taxonomic group: (**A**) parasitic diseases, (**B**) bacterial and fungal diseases, and (**C**) viral diseases.

**Figure 4 pathogens-10-00356-f004:**
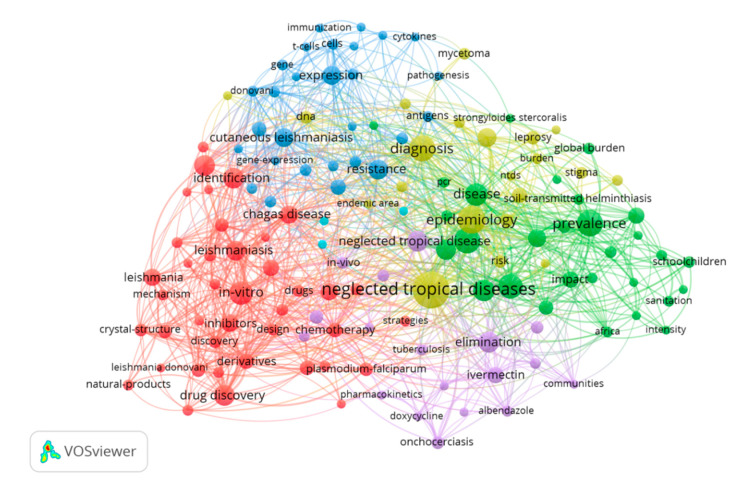
VOSviewer visualization of the most frequently related terms in retrieved articles regarding “neglected tropical diseases” over the period of 2010–2020 and extracted from the Web of Science (WoS). The size and color indicate the frequency and the cluster with which the terms have been appeared respectively. The closer two terms in the network the stronger their relation. The colors indicate clusters as assigned by VOSviewer.

**Figure 5 pathogens-10-00356-f005:**
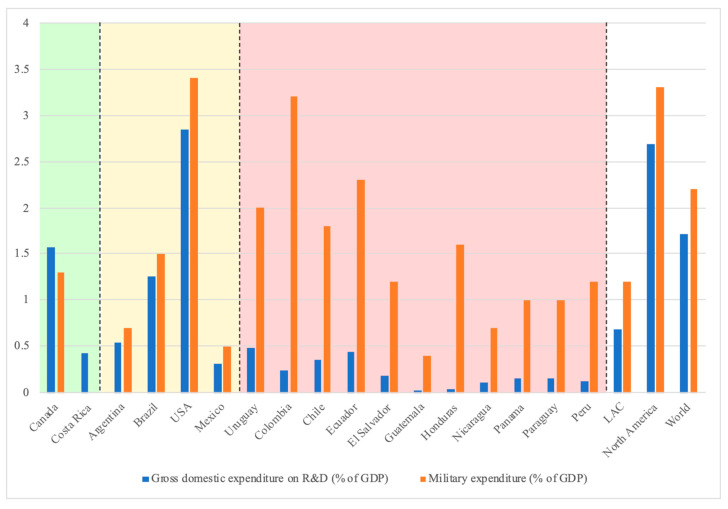
Gross domestic expenditure on research and development (R&D) as a percentage of the gross domestic product (GDP) (data taken from UNESCO Institute for Statistics), and military expenditure as a percentage of the GDP (data from The World Bank) of some American countries (the data reflects the information of the last year available). The countries within the green box have a higher percentage of investment in R&D than in military, the countries within the yellow box are those that resemble the world proportion, while the countries in the pink box are those in which there is a significantly higher expense in military than investment in R&D.

**Table 1 pathogens-10-00356-t001:** Number of scientific publications on 19 neglected tropical diseases and the “big-three” infectious diseases (malaria, tuberculosis and HIV/AIDS), top institutions, most prolific countries, journals and authors in the timespan from 2010 to 2020.

Disease	[MeSH Terms] and Boolean Operators	Number of Publications Worldwide	Most Productive Institution	Country with the Highest Number of Publications	Most Frequent Source	LAC Country with the Highest Number of Publications	Most Productive Author, Affiliation
Buruli ulcer	[Buruli ulcer] OR [Mycobacterium ulcerans]	769	The University of Melbourne	USA	*Plos Neglected Tropical Diseases*	Brazil	Gerd Pluschke, Swiss Tropical and Public Health Institute (Swiss TPH)
Chagas disease	[Trypanosoma cruzi] OR [Chagas]	13675	Universidade de Sao Paulo	Brazil	*Plos Neglected Tropical Diseases*	Brazil	Marcel Kaiser, Universitat Basel
Chromomycosis	[Chromomycosis] OR [Chromoblastomycosis]	365	Sun Yat-Sen University	Brazil	*Mycoses*	Brazil	G. Sybren de Hoog, Radboud University Nijmegen Medical Centre
Taeniasis/cysticercosis	[Taenia solium] OR [Taenia saginata] OR [Cysticercosis] OR [Cysticercus cellulosae]	1745	Universidad Nacional Autónoma de México	USA	*The American Journal of Tropical Medicine and Hygiene*	Mexico	Héctor H García, Universidad Peruana Cayetano Heredia
Dengue	[Dengue]	18186	Mahidol University	USA	*The American Journal of Tropical Medicine and Hygiene*	Brazil	Eva Harris, University of California, Berkeley
Zika	[Zika]	9074	Centers for Disease Control and Prevention	USA	*The American Journal of Tropical Medicine and Hygiene*	Brazil	Viroj Wiwanitkit, Dr. D. Y. Patil Medical College, Hospital & Research Centre, Dr. D. Y. Patil Vidyapeeth, Pune
Chikungunya	[Chikungunya]	4969	Institut Pasteur	USA	*Plos Neglected Tropical Diseases*	Brazil	Scott Weaver, UT Medical Branch at Galveston
Echinococcocosis/Hydatidosis	[Echinococcus granulosus] OR [Echinococcosis] OR [Hydatid cyst] OR [Hydatid disease]	4936	Xinjiang Medical University	Turkey	*Plos Neglected Tropical Diseases*	Argentina	Hao Wen, Xinjiang Medical University
Fascioliasis	[Fasciola hepatica] OR [Fascioliasis]	1516	Queen´s University Belfast	Spain	*Veterinary Parasitology*	Brazil	Ian Fairweather, Queen’s University Belfast
Leishmaniasis	[Leishmania] OR [Leishmaniasis]	16589	Universidade de Sao Paulo	Brazil	*PLos Neglected Tropical Diseases*	Brazil	Santhanam Sundar, Banaras Hindu University, Institute of Medical Sciences
Leprosy	[Leprosy] OR [Hansen´s disease] OR [Mycobacterium leprae]	4888	Universidade de Sao Paulo	Brazil	*Leprosy Review*	Brazil	Euzenir Sarno, Fundaçao Oswaldo Cruz
Lymphatic filariasis	[Lymphatic filariasis] OR [Wuchereria bancrofti] OR [Brugia]	2228	Liverpool School of Tropical Medicine	USA	*The American Journal of Tropical Medicine and Hygiene*	Brazil	Gary J. Weil, University of Washington School of Medicine
Mycetoma	[Mycetoma]	451	University of Khartoum	Sudan	*Plos Neglected Tropical Diseases*	Mexico	Ahmed Hassan Fahal, Mycetoma Research Centre
Onchocerciasis	[Onchocerca] OR [Onchocerciasis]	1271	Ministry of Health Democratic Republic of Congo	USA	*The American Journal of Tropical Medicine and Hygiene*	Brazil	Robert Leon Colebunders, Universiteit Antwerpen
Rabies	[Rabies] AND [Virus]	4920	Centers for Disease Control and Prevention	USA	*Plos Neglected Tropical Diseases*	Brazil	Charles E. Rupprecht, LYSSA LLC
Schistosomiasis	[Schistosomiasis] OR [Schistosoma]	7678	Swiss Tropical and Public Health Institute	USA	*Plos Neglected Tropical Diseases*	Brazil	Jürg Rg Utzinger, Universitat Basel
Soil-transmitted helminths	[Soil-transmitted helminths] OR [Ascaris lumbricoides] OR [Trichuris trichiura] OR [Ancylostoma duodenale] OR [Necator americanus] OR [Strongyloides stercoralis] OR [Geohelminth]	3222	Swiss Tropical and Public Health Institute	USA	*Plos Neglected Tropical Diseases*	Brazil	Jürg Rg Utzinger, Universitat Basel
Trachoma	[Trachoma] OR [Chlamydia trachomatis]	5337	The London School of Hygiene & Tropical Medicine	USA	*Sexually Transmitted Infections*	Brazil	Sheila K. West, Wilmer Eye Institute
Yaws	[Yaws] AND [Pertenue] AND [Treponema]	111	Universitat de Barcelona	USA	*Plos Neglected Tropical Diseases*	Brazil	Oriol Mitjà, Hospital Clinic Barcelona
Malaria	[Malaria] OR [Plasmodium]	48484	University of Oxford	USA	*Malaria Journal*	Brazil	Nicholas J. White, Nuffield Department of Medicine
HIV/AIDS	TS = [Human immunodeficiency virus] AND TS = (Autoimmune deficiency syndrome]	46293	University of California San Francisco	USA	*Plos One*	Brazil	David Charles Montefiori, Duke University School of Medicine
Tuberculosis	[Tuberculosis]	70139	University of Cape Town	USA	*Plos One*	Brazil	Ying Zhang, Johns Hopkins Bloomberg School of Public Health

**Table 2 pathogens-10-00356-t002:** Number and percentage of scientific publications by topic in Latin America and the Caribbean, the Americas and in the world, between 2010 and 2019.

Disease	Nº of Publications Worldwide	Nº of Publications from LAC (% of the World Production)	Nº of Publications from USA and Canada (% of the World Production)
Chagas disease	13,675	7748 (56.67)	2398 (17.54)
Tuberculosis	70,139	6384 (9.10)	18,721 (26.69)
Leishmaniasis	16,589	6265 (37.77)	3142 (18.94)
Malaria	48,484	4642 (9.57)	17,793 (36.70)
Dengue	18,186	4607 (25.33)	5476 (30.11)
HIV/AIDS	46,293	3047 (6.58)	24,386 (52.68)
Zika	9074	2674 (29.47)	4427 (48.79)
Chikungunya	4969	1474 (29.66)	1700 (34.21)
Leprosy	4,888	1460 (29.87)	994 (20.34)
Schistosomiasis	7678	1325 (17.26)	2104 (27.40)
Soil-transmitted helminths	3222	679 (21.07)	945 (29.33)
Cysticercosis	1745	615 (35.24)	381 (21.83)
Rabies	4920	594 (12.07)	1530 (31.10)
Hydatid cyst	4936	510 (10.33)	331 (6.71)
Fascioliasis	1516	392 (25.86)	164 (10.82)
Trachoma	5337	330 (6.18)	2265 (42.44)
Chromomycosis	365	179 (49.04)	35 (9.59)
Onchocerciasis	1271	142 (11.17)	601 (47.29)
Lymphatic filariasis	2228	136 (6.10)	923 (41.43)
Mycetoma	451	91 (20.18)	74 (16.41)
Buruli ulcer	769	16 (2.08)	117 (15.21)
Yaws	111	6 (5.41)	58 (52.25)

## Data Availability

Data is contained within the article and Appendix A.

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
