# Peer review of "Publication Trends in Neglected Tropical Diseases of Latin America and the Caribbean: A Bibliometric Analysis"

_pathogens, 2021, doi:10.3390/pathogens10030356_

Round 1
Reviewer 1 Report
General comment.
The authors should clarify in the Methods whether they consider differentially the articles from a single LAC country, or published by more than one LAC country, or articles published by one or more LAC countries in collaboration with a different region, for example USA, Europe, Japan.
Specific comments.
Line 2. The authors could consider modifying the title to read: ‘…diseases OF Latin…’
Line 95/96/Table 1. Journal titles should be in italic font.
Line 95. Should be re-worded to: ‘Plos Neglected Tropical Diseases (for TEN out of 19 NTDs)’.
Line 111-115. Should be re-worded to agree with the legend for Figure 1: ‘These three topics had an average annual growth even higher than TB (5.73%), malaria (2.29%), and HIV / AIDS (-0.82%). The topics with the highest average annual decrease were yaws (n=-8.24%), mycetoma (n=-2.45%) and chromomycosis (n=-1.18%) (FIGURE 1A). However, the average yearly number of publications for any of the "big-three" was significantly higher than for any of the NTDs analyzed (FIGURE 1B)’.
Line 232. Regarding Supplementary Table 1, is this novel data that was generated during this analysis? If it was, then it should be first mentioned in the Results Section.
Line 250. The cited references for Figure 5, as shown in the legend (#32, #42), should be cited in the text of the Discussion.
Table 2. The authors could consider moving the last column ‘Nº of publications worldwide (100%)’ to the first column position, and remove ‘(100%)’.
Figure 4. This figure is shown in the Results but it is not clear that it is mentioned again in the Discussion.
Author Response
Reviewer #1
General comment.
The authors should clarify in the Methods whether they consider differentially the articles from a single LAC country, or published by more than one LAC country, or articles published by one or more LAC countries in collaboration with a different region, for example USA, Europe, Japan.
- The following paragraph has been added to the methods section: “As in any other bibliometric analysis, the results of the WoS search include publications whose authors may belong to several institutions and to more than one country. Consequently, in many cases the results are a reflection of the collaboration of LAC scientists with countries in other regions, or of scientists from two or more LAC countries”.
Specific comments.
Line 2. The authors could consider modifying the title to read: ‘…diseases OF Latin…’
- The title has been modified as suggested: “Publication trends in neglected tropical diseases of Latin America and the Caribbean: a bibliometric analysis”.
Line 95/96/Table 1. Journal titles should be in italic font.
- All journal titles have been written in italics.
Line 95. Should be re-worded to: ‘Plos Neglected Tropical Diseases (for TEN out of 19 NTDs)’.
- Thanks. The sentence has been corrected.
Line 111-115. Should be re-worded to agree with the legend for Figure 1: ‘These three topics had an average annual growth even higher than TB (5.73%), malaria (2.29%), and HIV / AIDS (-0.82%). The topics with the highest average annual decrease were yaws (n=-8.24%), mycetoma (n=-2.45%) and chromomycosis (n=-1.18%) (FIGURE 1A). However, the average yearly number of publications for any of the "big-three" was significantly higher than for any of the NTDs analyzed (FIGURE 1B)’.
- The paragraph has been reworded as suggested.
Line 232. Regarding Supplementary Table 1, is this novel data that was generated during this analysis? If it was, then it should be first mentioned in the Results Section.
- The supplementary table 1 has been mentioned in the section 2.5 of the Results. Thanks for noticing.
Line 250. The cited references for Figure 5, as shown in the legend (#32, #42), should be cited in the text of the Discussion.
- Both references have been removed from the legend and have been included in the Discussion.
Table 2. The authors could consider moving the last column ‘Nº of publications worldwide (100%)’ to the first column position, and remove ‘(100%)’.
- Done.
Figure 4. This figure is shown in the Results but it is not clear that it is mentioned again in the Discussion.
- The reviewer is right. The output of the VOSviewer software is only a visual aid to help the reader understand which are the most used terms in relation to the topic of NTDs. We do not consider it relevant to discuss this topic in order not to dilute the discussion and to be able to focus on other results with greater impact.
Reviewer 2 Report
See the attachmnet.

Author Response
Reviewer #2
The authors present the first bibliometric analysis to assess the publication trends in endemic Neglected Tropical Diseases in Latin America and the Caribbean in scientific documents. However, according to the authors (lines 332-333) they present in their article only a reflection of reality due to the estimable limitations of their study
(lines 325-333). In order not to be a diffuse reflection and to make the study more accurate, the MS must be considerably improved.
Line 77. 2.1. Publication output. Specify the total number of publications found considering all the diseases analysed.
- The following text has been included as lead sentence to section 2.1: “The bibliographic search on the 19 NTDs described above, yielded a total of 266,846 publications worldwide for the period 2010-2019”.
Line 103. Table 1: General: The table shows information on the "Most productive institution", "Country with the highest number of publications" and "Most productive author and affiliation". Nothing is commented about institutions or authors in the main text. For example, curious cases like Buruli ulcer, in which the most productive institution is an Australian University, the country with the highest number of publications is the USA, and the most productive author is a Swiss researcher working in Switzerland.
- The results shown in Table 1 reveal interesting information, such as that indicated by the reviewer in the case of buruli ulcer, for which the institution does not coincide with the country or origin of the most prolific author. It is true that nothing is mentioned in the text about the institutions or the authors. Due to vast amount of information obtained in a bibliometric search, authors must be selective in choosing to discuss the elements that are most relevant to the central objective of the study. In our work, manuscript, data with regards to authors and institutions are only described in a table, so that interested readers can have a quick access. The style of data description used in our manuscript closely adheres to the style of similar bibliometric publications.
- Column 1: Above the NTD column, “Disease” is missing. In addition, it would be clearer if the diseases were ordered according to the number of documents worlwide, i.e. from the highest number of documents to the lowest one.
- The word “Disease” has been included as a heading in the first column of Table 1. We agree that there are many ways to organized data in a table. While the order suggested by the reviewer is attractive, we determined that an alphabetical order would provide the reader a finding strategy aligned with the study objective. In addition, the three non-NTDs have been placed at the bottom of the table to separate them from the rest.
- Column 1: Echinococcosis is written Echinocococosis, correct it.
- Corrected. Thanks a lot.
- Column 3: Is “documents” synonymous with “publications”? “Documents” appear only in column 3, whereas “publications” appear in the rest of columns in both, tables 1 and 2.
- The word "documents" has been replaced by "publications" to homogenize the terms.
Table 2: - Column 1: Order the NTDs according to the number of publications from LAC, from the highest number to the lowest one.
- The list of diseases has been ordered according to the number of LAC publications.
- Column 2: check % of Chagas disease according to its true number of publications worldwide (13675 in Table 2, however 9253 in Table 1).
- We really appreciate noticing the error in the table. We have made the correction.
- Column 4 is unnecessary (it is only the addition of 1 + 2) as the article specifically refers to LAC.
- We have removed column 4 from table 2.
- Column 5: Check Chagas disease Nº of publications worldwide since in Table 1 it says 9253 but in Table 2 it says 13675.
- Corrected. Thanks.
Figure 3 A: try to improve the colourings in order to better differentiate the 10 parasitic diseases.
- We appreciate the observation but as the main objective of Figure 3.A. is to show that the number of publications on malaria is much higher than that of other parasitic diseases. Since the lines of the other 9 diseases shown are so close together in the graph, we didn’t think it would be helpful to change the colors.
Lines 170-182: These lines are a background on NTDs, therefore they should be part of the Introduction section.
- We appreciate the observation. We have reformulated the introduction, adding the first paragraph of the discussion there.
Lines 185-186: This is not entirely true. Only when different NTDs have similar incidence, morbidity and mortality, is possible to affirm that certain NTDs are more neglected than others. If this is not the case, the lower number of publications on certain NTDs might be is related to their lower incidence, morbidity and mortality. Logically, more severe or prevalent diseases deserve more attention.
- To avoid potential controversy around by the phrase "more neglected than others", the phrase has been reformulated as follows: “From a scientific perspective, it appears that some NTDs have received more attention than others”.
Lines 203-207: The fact that the USA had the largest output of scientific publications on Tropical Medicine (or in many other subjects) in the world was obvious and already known without carrying out any bibliometric study. The USA, with more than 331 million people, has states larger than most LAC countries.
- Certainly, the United States has contributed more to scientific outputs in the last century than any other country in the world. Notwithstanding, bibliographic studies are still needed to ascertain trends (see, for example https://ncses.nsf.gov/pubs/nsb20206/publication-output-by-region-country-or-economy). The present manuscript aimed to determine the contribution of the LAC countries regarding the NTDs. With USA being part of the Americas, we would be remiss if we didn’t take their contribution into account. We deem it important to highlight that in spite the fact that most NTDs are not autochthonous to USA, this nation still has a greater production of scientific publications that LAC.
Lines 207-210/232-233: There does not have to be a relationship between the disease studied and the country of origin of the publication. It would be interesting to analyse the country origin of the material studied and/or the study areas in the publications elaborated, for example, in the USA. For sure, the studied materials, and/or the studied areas, belong to LMICs. In fact, for example, as far as I know, most of the material studied in the publications of Fasciola in Spain came from South American, African or Asian Countries. And I can assure the authors of the MS that neither political nor military interests prompt those researchers to study Fascioliasis
in endemic countries. The authors of the article seem to agree with Palmblad et al. when "accusing" scientists from HICs of being prompted by political or military interests when establishing international cooperation with LICs to study diseases not endemic in HICs. I firmly disagree with that vision of the current International Cooperation Programmes for Research and Development or the current Eradication Campaigns such as those for eradicating Malaria, Dracunculiasis, Lymphatic Filariasis, Onchocerciasis, etc.
- The suggestion of analyzing studies by the biological samples’ provenance is an interesting one. So far, we have not seen such study -at least in the frame of scientific output- and we hope that our work will inspire future research in this regard. With regards to the reviewer’s position on Palmblad et at (2017), we are simply offering potential explanations to the observed disparities. We ourselves publish frequently as part of international collaboration solely motivated by the advancement of knowledge but at the same time we cannot deny that historic and geopolitical factors do play a role in scientific research.
Lines 248-258/263-270: It is unnecessary and useless in the article to deal with military spending and research budgets. The WHO acknowledges that there are only three diseases that are genuinely ‘neglected’: African trypanosomiasis, Leishmaniasis and Chagas disease. A large proportion of illnesses in LMICs are entirely avoidable or treatable with existing drugs or interventions. Most of the disease burden in these countries is rooted in the consequences of poverty, such as poor nutrition, lack of access to proper sanitation and health education. Therefore, in the article, it would be better to encourage those countries to start investing, first of all, in sanitation and education, and then in scientific research. That achievement will lead to others.
- We aim to point out that a shift in countries’ priorities towards more research and development (which will also have great returns in societal health and well-being) would be a desirable change of perspective. This is particularly true in LAC countries with little to no risk of international armed conflict. If fact, as the reviewer indicates, if countries invested in feasible interventions, the burden of NTDs would decrease considerably. In our manuscript we postulate, as an inference to Figure 5, that low- and middle-income countries need to invest more resources in sanitation and education and then in scientific research.
Lines 278-285: Apparently, the searched terms used by the authors were exclusively those appearing on the Medical Subject Headings website. Therefore, the following parasitological terms (frequently used by authors) must also be included in the search: American Trypanosomiasis; Taeniasis, Taeniosis; Cystic hydatidosis; Fasciolosis; Leishmaniosis, Wuchereriasis, Onchocercosis; River blindness; Bilharziasis; Ascariasis, Trichuriasis, Ancylostomiasis, Necatoriasis, Strongyloidiasis. Likewise, terms in Spanish must also be included, such as Tripanosomiasis americana, Cisticercosis, Equinococosis, Hidatidosis, Filariasis linfática, Oncocercosis, Ceguera de los Rios, Esquistosomiasis, Enfermedades transmitidas por el suelo, Geohelmintos, Paludismo.
- The reviewer suggests that other terms be added in the searches to have a more complete picture of reality. In fact, the greater the number of terms used in the search, the greater the number of accessions returned by the database. We found this suggestion interesting and carried out the exercise but found negligible differences with our search strategy, as follows: for Chagas disease, less than 0.4% more publications were obtained; for cysticercosis/taeniasis, less than 2% of new publications were obtained; for leishmaniasis, less than 0.5% of new publications were obtained. Moreover, when terms in Spanish were used, the number of publications did not increase in most cases. As we have acknowledged in the limitations, no bibliometric analysis should be considered a perfect reflection of reality. Nevertheless, selecting appropriate MeSH terms is crucial to obtaining credible and replicable findings that fulfill the objectives established for the study.
Lines 320-324: Conclusions is an optional section, so delete it. I do not believe the obtained results (quite predictable, by the way) induce any decision in researchers or policy makers in order to achieve the SDGs for global health, above all taking into account the considerable limitations of their work pointed out by the authors.
- We respect the standpoint of the reviewer but believe that our findings as well as those from similar work may have an impact in helping decision-makers at the local, national or transnational level. Further, even when results are “predictable” it is essential to provide evidence in some cases repeatedly, to effect change. Whether or not decisions are made in face of any evidence is beyond researcher’s power.
- Limitations. When limitations can be easily overcome (only investing more time than the authors did), these must, obviously, be tackled. More than one database must be analysed as well as the additional aforementioned terms of certain parasitic diseases (and for the rest of the diseases analysed too) must be included in the bibliometric analysis.
- Like any type of study, bibliometric analyzes also have limitations. Limitations are not synonymous with error. The term “limitation” indicates that the scope of any study does not cover the entire universe. Sometimes, like the current manuscript, having selected only WoS as the database, leaving out others (like PubMed), does not indicate that the results are incorrect. Since the databases are interconnected, the vast majority of the articles would be common to all of them. WoS was selected as the only source of consultation because is the most cited database and our study included too many variables (22 diseases, all the countries of the Americas, >10 analysis criteria), so including two or more databases would have made the document difficult to read. Either way, the number of similar articles in the literature that base their analysis on a single database (WoS) is staggering.
Reviewer 3 Report
Although the data presented is interesting and bring up important observations, there are implicit bias that may affected the results. Many of these implicit bias were well described by the authors; however, the analysis of more sources of references would reduce the risks. Consider adding data or justification.
Additionally, did the authors consider explore resources in the native language of the Countries? If so, would be best to add the information to the manuscript. If not, please considered add discussion about this potential implicit bias of the data.
Please consider to include justification/explanation for choosing to restrict the bibliometric analysis to WoS.
Author Response
Reviewer #3
Although the data presented is interesting and bring up important observations, there are implicit bias that may affected the results. Many of these implicit bias(ES) were well described by the authors; however, the analysis of more sources of references would reduce the risks. Consider adding data or justification.
- Like any type of study, bibliometric analyzes also have limitations. Limitations are not synonymous with error. The term “limitation” indicates that the scope of any study does not cover the entire universe. Sometimes, like the current manuscript, having selected only WoS as the database, leaving out others (like PubMed), does not indicate that the results are incorrect. Since the databases are interconnected, the vast majority of the articles would be common to all of them. WoS was selected as the only source of consultation because is the most cited database and our study included too many variables (22 diseases, all the countries of the Americas, >10 analysis criteria), so including two or more databases would have made the document difficult to read.
Additionally, did the authors consider explore resources in the native language of the Countries? If so, would be best to add the information to the manuscript. If not, please considered add discussion about this potential implicit bias of the data.
- Only English search terms were used because, in general, terms in other languages ​​such as Spanish or Portuguese do not significantly increase the results. This is because articles published in other languages ​​use English keywords in all indexed journals.
Please consider to include justification/explanation for choosing to restrict the bibliometric analysis to WoS.
- While it is true that some publications with narrow objectives, (e.g. analysis of a single disease, or of a single country), often use more than one database; however, most of the more complex bibliometric analyses use only one as database because the databases are interconnected and yield approximately the same information. WoS is a good option; it is universally accepted as the most reliable database for this type of analysis because it allows fewer false positives in relation to other search engines.
Reviewer 4 Report
I think the article is simple, straightforward and well written.
Despite the significant increase in the recent years, of interest by the scientific community, the number of NTD publications is much smaller than the ones focusing to malaria or HIV and this can be a push for researches in the area of communicable diseases to deserve more attention to NTD.
If I can give a small suggestion to the authors I would mention in the introduction that the burden of disease estimated for the 20 NTD is on the same order of magnitude than the one of malaria, TB or HIV. (Please see the graph at https://vizhub.healthdata.org/gbd-compare/ ) and therefore that the different in term of research and publication is not justified.
Author Response
Reviewer #4
I think the article is simple, straightforward and well written.
Despite the significant increase in the recent years, of interest by the scientific community, the number of NTD publications is much smaller than the ones focusing to malaria or HIV and this can be a push for researches in the area of communicable diseases to deserve more attention to NTD.
If I can give a small suggestion to the authors I would mention in the introduction that the burden of disease estimated for the 20 NTD is on the same order of magnitude than the one of malaria, TB or HIV. (Please see the graph at https://vizhub.healthdata.org/gbd-compare/ ) and therefore that the different in term of research and publication is not justified.
- We welcome the suggestion. The following sentence has been included in the introduction: “although the collective burden of disease from NTDs was —and still is— higher in some regions of the world, such as Latin America and the Caribbean (LAC)”.
Round 2
Reviewer 2 Report
See attachment.

Author Response
March 12th, 2021
Pathogens
Dear Editor:
Regarding the manuscript entitled "Publication trends in neglected tropical diseases in Latin America and the Caribbean: a bibliometric analysis", we have received the second round of observations suggested by the reviewer #2. We appreciate all the suggestions received to improve the scientific quality of the manuscript.
Below you will find the point-by-point responses to all the comments received and a new version of the document using the track changes tool.
Sincerely,
Gustavo A. Fontecha S., PhD.
Reviewer #2 (second round)
“I am only commenting my discrepancies with authors’s answers. The rest is OK.
- Authors: The results shown in Table 1 reveal interesting information, such as that indicated by the reviewer in the case of buruli ulcer, for which the institution does not coincide with the country or origin of the most prolific author. It is true that nothing is mentioned in the text about the institutions or the authors. Due to vast amount of information obtained in a bibliometric search, authors must be selective in choosing to discuss the elements that are most relevant to the central objective of the study. In our work, manuscript, data with regards to authors and institutions are only described in a table, so that interested readers can have a quick access. The style of data description used in our manuscript closely adheres to the style of similar bibliometric publications.
Reviewer#2: I am not saying that the information on discrepancies among institutions, countries and authors must be included in the Discussion section. However, that information should be commented on Results, the same as other results have been commented on different subsections (2.1-2.6). Concerning
style, tables must not repeat information provided in the main text. However, all the information appearing in the tables must be commented on, even if only in a short sentence, in the main text.
Line 126 (new version): Table is written with small letter. Change it to capital
letter.”
Response from authors to second round comments:
(a) Section 2.3 has been renamed 2.3. “Most productive journals, languages, document types, most prolific authors and institutions”
The paragraph has been modified as follows: “When considering the 19 NTDs as research topics, the University of Sao Paulo in Brazil was the most productive institution for Chagas disease, Leishmaniasis and Leprosy, while the Swiss Institute of Public and Tropical Health ranked first for publications on Schistosomiasis and Soil-Transmitted Helminthiases, The US Centers for Disease Control and Prevention (CDC) contributed with the largest number of publications for Zika and Rabies.
In terms of authors’ affiliations, institutions in the United States and Switzerland had the highest output for five and four of the 19 NTDs, respectively.
The analysis by type of institution, results indicated that for 13 NTDs most productive authors were affiliated with universities, whereas the rest were hospitals, public health institutes, research centers, or foundations”.
(b) The word Table is capitalized throughout the document.
Reviewer #2 (second round)
- Authors: the suggestion of analyzing studies by the biological samples’ provenance is an interesting one. So far, we have not seen such study -at least in the frame of scientific output- and we hope that our work will inspire future research in this regard. With regards to the reviewer’s position on Palmblad et at (2017), we are simply offering potential explanations to the observed disparities. We ourselves publish frequently as part of international collaboration solely motivated by the advancement of knowledge but at the same time we cannot deny that historic and geopolitical factors do play a role in scientific
Reviewer#2: The authors do not offer potential explanations in the article, they agree with: (lines 209-210) research outputs on tropical medicine are related to historical events, or even to current political or military interests. This sentence should be changed or clarified by means of specifying examples of countries that, currently, are moved by political or military interests when cooperating with LICs to control/eradicate human diseases. If the authors denounce or agree with that, they should provide the information on the countries they know are currently acting that way.
Response from authors to second round comments:
We regret the misunderstanding. By “political or military interest” we did not intend to denounce dark motives for such collaborations. To clarify, we have reworked that section as follows:
Historically, these collaborations, as noted by Palmblad and Torvik 2017 may stem from shared colonial history and language [19]. In present times, incipient research development in many tropical countries drives the need for scientific collaboration with more developed nations. In addition, research outputs in this field maybe reflect common interest related to geopolitical circumstances. An example of the latter is the fact that the U. S. Army and the U. S. Navy deploy temporary or permanent units to developing countries (e.g., in the Americas: Peru, Colombia, and Honduras, among others) to participate in peacekeeping operations, joint military exercises, and establish collaborations on epidemiological surveillance with local health or academics institutions [28]. Since military personnel can either spread infectious diseases within a population or become at risk of infection by pathogens endemic to the host tropical countries, collaborative biomedical and epidemiological research becomes indispensable to address infectious diseases risks.
Reviewer #2 (second round)
Authors: The reviewer suggests that other terms be added in the searches to have a more complete picture of reality. In fact, the greater the number of terms used in the search, the greater the number of accessions returned by the database. We found this suggestion interesting and carried out the exercise but found negligible differences with our search strategy, as follows: for Chagas disease, less than 0.4% more publications were obtained; for cysticercosis/taeniasis, less than 2% of new publications were obtained; for leishmaniasis, less than 0.5% of new publications were obtained. Moreover, when terms in Spanish were used, the number of publications did not increase in most cases. As we have acknowledged in the limitations, no bibliometric analysis should be considered a perfect reflection of reality. Nevertheless, selecting appropriate MeSH terms is crucial to obtaining credible and replicable
findings that fulfill the objectives established for the study.
Reviewer#2: According to these comments, less than 0.4% (for example, 0.35%), of the Chagas disease publications (13,675) means that about 48 publications have not been considered, or about 30 in the case of Taeniasis/Cysticercosis or 58 in the case of Leishmaniasis. Therefore, more than 130 new publications (considering only these 3 of the more than the 20 diseases analysed), have not been included in the analysis. The new findings for all the new terms used should be included.
Reviewer: Limitations: The authors have maintained the same limitations as those in the first version. I would understand one limitation, not two. I mean: either one database with more terms (like those already suggested), or more than one database with the limited terms the authors used. Only one database, although representative, with these limited search terms is, from my point of
view, not acceptable, as I already commented on in the first version of the MS.
Response from authors to second round comments:
We fully understand the reviewer's point of view that including new keywords in languages ​​other than English, or a second database, would provide a closer look at reality.
However, we are unable to satisfy the recommendation as it would entail a significant change in Methods and Results (recalculating statistics, rewriting tables and redoing graphs) as well. Such change would lead to a significantly changed manuscript, which would have to be reviewed de novo by the 3 other reviewers —thus restarting the entire process).
We trust that Reviewer #2 concedes that our work aimed at revealing publication TRENDS in in the NTD field in the specific region of Americas.
While following the recommendation would in fact yield more results and increase absolute numbers in our findings, we posit that exercise would not modify the trends shown, nor would it result in noteworthy differences in the most productive institutions, countries or authors.
We regret not being able to accommodate the suggestion on this occasion.